# *LazyFrog*: Advancing Security and Efficiency in Commercial Wireless Charging with Adaptive Frequency Hopping

**DOI:** 10.3390/s24082571

**Published:** 2024-04-17

**Authors:** Sungkyu Ahn, Hyelim Jung, Ki-Woong Park

**Affiliations:** 1SysCore Lab., Sejong University, Seoul 05006, Republic of Korea; vh2000xm@sju.ac.kr (S.A.); hyello13@gmail.com (H.J.); 2Department of Information Security, and Convergence Engineering for Intelligent Drone, Sejong University, Seoul 05006, Republic of Korea

**Keywords:** wireless charge, charging attack, dynamic charging, frequency hopping

## Abstract

With the proliferation of electronic devices and electricity-based mobility solutions, the significance of wireless power transfer technology has increased substantially. However, ensuring secure and reliable power transmission to authorized users remains a significant challenge. Addressing this complex issue requires an integrated approach that balances efficiency, stability, and security considerations. While current efforts primarily focus on improving charging efficiency and user convenience, integrating robust security measures into wireless charging infrastructure is challenging due to its inherently open nature and susceptibility to external interference. Technical advancements are required to strengthen the security of the wireless charging infrastructure; however, these should be balanced with power loss management. This study tackles two core issues: the increasing hardware requirements for billing system authentication protocols and the interception of wireless charging signals by unauthorized users, leading to power theft and subsequent losses. To address these challenges, we propose a mechanism termed “*LazyFrog*”. This mechanism dynamically adjusts the frequency hopping schedule, activating frequency changes only in response to detected threats during remote charging or upon identifying unauthorized access attempts. The proposed mechanism compares the expected power reception at the device with the actual power supplied by the charging station, enabling the detection of abnormal power losses. By minimizing unnecessary frequency changes and optimizing energy consumption, *LazyFrog* reduces hardware requirements. Moreover, we have implemented a relative distance estimation mechanism to facilitate efficient power transfer as wireless devices move within the charging environment. With these features, *LazyFrog* demonstrates a secure, flexible, and energy-efficient wireless charging system ready for practical application.

## 1. Introduction

The proliferation of electronic devices and electricity-based mobility solutions is a significant trend in modern society, and the operational functionality and sustainability of such devices are becoming increasingly critical [1,2]. In this context, the role of the charging infrastructure is immensely significant, particularly that of convenient and efficient wireless charging technology, which has become integral to the daily lives of users [3,4,5].

Wireless charging technology can be divided into three types based on distance. The first type, i.e., short-range charging, involves magnetic inductive coupling and can wirelessly charge mobile devices at short distances. Although short-range charging is generally efficient, the charging efficiency significantly decreases when the distance exceeds 40 mm [6]. Hence, users cannot move while using short-range charging technology. The second type is mid-range charging, which can transmit power wirelessly to distances ranging from 1.0–4.5 m. The third type is long-range charging, which is capable of transmitting power wirelessly up to distances of 15 m. Multiple commercial wireless charging services support middle-range and long-range wireless charging [7].

Commercial mid-range wireless charging technologies include AirFuel RF and Energous WattUp. AirFuel RF uses radio frequency to transmit power over mid-range from centimeters to several meters, thus providing wireless charging services capable of charging multiple devices [8]. It can be applied to wearable devices and smart home appliances. The Energous WattUp technology employs radio frequency to charge multiple devices simultaneously across various distances from short to long range [9]. It is applicable to wearable devices, healthcare equipment, smart home and Internet of Things (IoT) devices, and office equipment, with a focus on small electronic devices and wearables, and it was showcased at the Consumer Electronics Show (CES) 2023.

However, wireless charging technologies are subject to significant security challenges [10,11,12]. Specifically, the task of transmitting power accurately and safely to authorized users or devices is complex [13,14]. Attackers can exploit power leaks that enable the charging of unauthorized devices, thereby affecting the charging of other devices and leading to critical operational problems. These security vulnerabilities can undermine the reliability of wireless charging technologies and threaten user security [15,16].

Numerous services prevent frequency interference via security measures by delivering power to specific points or encrypting frequencies to transmit power only to authenticated devices. Extensive research has been conducted to ensure that only authenticated users can use the charging service. Zhang et al. [17] used chaos theory to generate different frequencies, thus enabling frequencies to appear randomly. They established a secure energy transmission channel that prevents unauthorized receivers from stealing power. Only users with knowledge of the decryption key can identify the frequency and charge their mobile devices. The Massachusetts Institute of Technology (MIT) used elliptic curve cryptography authentication to identify unauthorized users [18].

However, the implementation of such security technologies in wireless charging is generally less efficient owing to its complexity. To overcome these limitations, numerous commercial products have adopted frequency hopping techniques to enhance security [19,20]. This technology changes the charging frequencies, thereby significantly enhancing the security of wireless charging services.

Figure 1 illustrates the frequency-scheduling method. It can be assumed that a charging system changes its frequency five times per second. For example, if an unauthorized user is aware of a specific frequency f2, the system limits the battery charging of the unauthorized user to a maximum of 20%. As the frequency range expands, the charging efficiency for unauthorized users tends to decrease. However, for pay-as-you-go wireless charging services, the delay when generating a frequency schedule for changing frequencies is a significant problem. Synchronization between mobile devices and wireless chargers should be established to change the frequency. However, delays inevitably occur when mobile devices and wireless chargers are synchronized using a network time-protocol (NTP) server [21]. The delay range varies from milliseconds to tens of milliseconds. Mobile devices are required connect to an NTP server via the Internet; therefore, the delay is up to tens of milliseconds.

To address these issues, an integrated approach that focuses on efficiency, economy, convenience, and stability is required. The development of technologies that enhance security while minimizing energy loss is critical. Concurrently, cost-effective solutions are required to ensure the long-term sustainability of charging systems. Finally, charging system designs that consider both maintenance costs and security threats are required to improve the proliferation and stability of wireless charging technology.

Therefore, we herein propose the *LazyFrog* mechanism, which regulates access and minimizes channel switching frequency to reduce power leakage, thereby providing legitimate users with a secure power supply. This mechanism includes a situational awareness mechanism to detect and respond to security threats that may arise during wireless charging. Additionally, the situational awareness mechanism minimizes the communication frequency between terminals and charging stations, thus enhancing efficiency while ensuring that physical threats to specific devices do not affect the entire system.

*LazyFrog* aims to enhance the availability of wireless charging via three mechanisms. First, the Patrol mechanism guides the frequencies that provide stable charging without unauthorized users via periodic power transmission tests between charging stations (APs). Second, the AP allocation mechanism shares charging information among the APs surrounding a wireless charging receiver in real-time and efficiently assigns APs to the receiver. Third, the detection and hopping mechanism provides stable charging by detecting unauthorized charging users and changing frequencies by monitoring charging power in real-time via the AP management server.

Figure 2 illustrates the operational environment of the wireless charging service structured to explain the three mechanisms of *LazyFrog*, and the scenarios in which each mechanism operates. Part (0) of Figure 2 represents the wireless charging service environment developed by *LazyFrog*, which consists of multiple APs and a server, with users owning electronic devices that utilize wireless charging while moving around. The APs execute the *LazyFrog* mechanisms of patrol, AP allocation, detection, and hopping. The server holds information to manage the device details of users and the power transmission frequencies used by the APs.

Part (1) of Figure 2 outlines the operational scenario of the *LazyFrog* patrol mechanism. The APs (AP1, AP2, and AP3) continuously perform wireless power transmission among themselves using each frequency existing within the available frequency range, thus securing safe frequencies and unauthorized access devices in the process. The server manages the list of safe frequencies transmitted from the APs. If an attacker gains unauthorized access to a frequency to receive power, the AP and user device implement a defense strategy (detection and hopping) by jumping to a safe frequency, for which the server updates the list of safe frequencies for the AP and sets up the user to access the safe frequency.

Part (2) describes the scenario for explaining the operation of the *LazyFrog* AP Allocation mechanism, which evaluates the received signal strength indicator (RSSI) from various APs to determine the most suitable AP for the device. For example, a device initially connected to AP1 may be reassigned to AP2 by the server if AP2 provides a superior RSSI value as the user moves. This process ensures an efficient and stable charging by providing the user with an appropriate AP suggestion with safe frequency information, thus allowing authorized users to connect their devices for wireless charging.

Part (3) explains the scenario for describing the operation of the *LazyFrog* detection and hopping mechanism, which actively monitors the power loss received by user devices, which is indicative of unauthorized devices attempting to receive power. If a discrepancy between the estimated power reception by the user device and actual power received from the AP is detected, the system starts to hop to a safe frequency collected by the patrol mechanism to mitigate potential interference and maintain a safe charging operation. The user device receives the changed frequency information from the server and changes its receiving frequency, thus accordingly ensuring safe charging.

The remainder of this paper is organized as follows: Section 2 reviews related work and existing charging systems. Section 3 provides an overall summary of the *LazyFrog* mechanism and its three components (devices, charging stations, and servers) and describes each stage of the mechanism. Section 4 evaluates the performance of the proposed *LazyFrog* mechanism, and Section 5 presents the conclusions.

## 2. Related Work

Security vulnerabilities in wireless charging infrastructure can enable unauthorized power use, which risks the reliability of wireless charging technology and user security. Addressing these issues requires an integrated approach that focuses on efficiency, economy, convenience, and stability. Moreover, stable and effective security should be provided by integrating security techniques such as frequency hopping while minimizing power loss.

This section describes previous studies related to the charging frequency interference of wireless charging technology, which forms the basis of the *LazyFrog* mechanism proposed in this paper. In addition, a discussion is presented on previous research focused on enhancing the availability of charging power for wireless charging technologies.

Previous studies contributed technical advancements to enhance the power availability of wireless charging technologies, including hardware improvements for higher power efficiency and software-based algorithm enhancements.

Elghanam et al. [22] researched the optimization of high-power, high-efficiency, and error-tolerant dynamic wireless charging (DWC) systems for electric vehicles (EVs), considering road and vehicle specifications. Resonant operation and maximum power transmission efficiency were achieved by tuning the inductor–capacitor–capacitor (LCC) compensation components, improving the design of the inductive links, and implementing closed-loop control systems to track the maximum power operation point under various coupling conditions. This compensated for the energy consumed while in motion and maintained an average power transmission efficiency of over 90%, thus providing an excellent lateral misalignment tolerance of ±200 mm. Wu et al. [23] aimed to address the high power and efficiency challenges of traditional wireless power transmission (WPT) systems as well as their wide tolerance for misalignment. They proposed a novel high-order WPT system utilizing parity-time (PT) symmetry-based multiple separated receiving coils, which maintains constant output power and transmission efficiency regardless of variations in the coupling coefficient. This complex design involves multiple separated receiving coils interacting with a transmitting coil, enhancing the system’s robustness to changes in the coupling coefficient, and thereby maintaining constant output power and efficiency even in the presence of misalignment between transmitting and receiving coils. Negative resistance was introduced using semiconductor switching devices and control circuits to regulate the input power, ensuring that the current and voltage of the transmitting coil always maintain the same phase angle. Experimental results showed that this system, thanks to the efficiency improvements from using multiple separated receiving coils, maintained stable power transmission with a constant efficiency of 96.1% over an air gap ranging from 100–200 mm. Moreover, the power transmission efficiency was further improved by the magnetic coupler design and by incorporating negative resistance. Tavakoli et al. [24] addressed cost efficiency and performance optimization of EV charging systems via dynamic wireless power transmission (DWPT), and proposed methods to mitigate the high costs and efficiency challenges associated with DWPT, particularly focusing on the expense of ground assemblies (GAs) installed on roads. A cost-efficiency optimization algorithm was developed to optimize the design of the DWPT transmitter (Tx) pads, maximize their efficiency, and minimize the cost of the GAs. In the study, a system cost model involving coil winding position optimization was developed, considering the lateral misalignment of the EV. In the performance evaluation, the optimized 3.7-kVA Tx pad was tested under various operating conditions, thus achieving a statistically expected efficiency of 96%, a GA cost of 1004 per meter, and an optimal length of 1.75 m.

Other studies improved the power efficiency of wireless charging technology by enhancing the associated algorithms. Gharaei et al. [25] proposed methods to improve energy efficiency and network lifespan by optimizing the moving path of wireless mobile chargers (WMCs) in response to the energy constraint issue of wireless sensor networks. These methods are focused on the adaptability of WMCs, which are designed to efficiently navigate toward and charge the dynamically located devices within the network, thus ensuring optimal energy distribution and prolonged network operation. Two algorithms have been proposed that optimize the moving trajectory of the WMC and consider the balanced energy depletion time of the sensor nodes. The rechargeable optimized wireless mobile charge (ROWMC) algorithm optimizes the moving path of the WMC to minimize the remaining lifespan dispersion of the sensor nodes and to improve the overall lifespan of the network. The objective of the charging time-optimized wireless mobile charge (CTOWMC) algorithm is to maximize the network lifespan by charging sensor nodes using the WMC when the remaining energy of the node falls below a threshold. In the numerical performance evaluation, the ROWMC algorithm improved the network lifespan by reducing the remaining lifespan dispersion when compared with other related algorithms, and the CTOWMC algorithm extended the network lifespan in low-density scenarios, thus reducing the level of transmitted energy and decreasing overhead. Jin et al. [26] proposed algorithms to address issues related to EV charging, particularly high-charging station installation costs, low-charging efficiency, and voltage variations in the power grid, which hindered the popularization of EVs. They proposed an EV wireless charging system utilizing urban bus networks, OnLine Electric Vehicle (OLEV) systems, and microwave power transfer (MPT) technology. The bus network-assisted wireless charging EV Route Scheduling (BRS) problem was defined, an approximate route scheduling algorithm (RSA) was proposed to solve it, and a bus network-assisted conflict-free EV route scheduling (BFRS) problem was developed to prevent charging schedule conflicts and alleviate traffic congestion. They also proposed a conflict-free route-scheduling algorithm (FRSA). In the performance evaluation against existing solutions, the RSA and FRSA achieved average remaining energy increases of 67.66% and 50.36%, respectively. The RSA reduced the average travel time by 22.22% and retained 77.23% of the energy, and the FRSA retained 83.51% of the energy but with an average additional travel time of 3.62%.

Hardware-improved wireless charging technologies focus on reducing electrical leakage, thus increasing charging efficiency, improving coil efficiency, and minimizing energy loss during power transfer. These improve the amount of charge and overall charging system efficiency. In contrast, wireless charging technology has been improved via software and algorithms to detect and efficiently manage power leakages caused by algorithms, in order to optimize power transmission. This results in a higher charging efficiency when compared with traditional charging methods.

The traditional approaches described in this section are limited. Hardware improvements increase the cost of wireless charging transmitters and incur higher production and maintenance expenses. Although software and algorithm improvements reduced the power leakage caused by device movement, the approach proposed in this paper performs AP searches that can provide high-efficiency power to devices despite device movement and implements algorithms to diminish the threat of power reductions due to unauthorized user access, thereby providing high-efficiency power to wireless charging receivers.

## 3. Design of *LazyFrog*

This section introduces the *LazyFrog* mechanism design and its key functionalities. Moreover, it details the threats to the charging system and presents a secure charging mechanism for addressing these challenges. The interactions and data flow among the core components of the charging system, namely, devices, charging stations, and servers, are detailed. In particular, this section emphasizes approaches to handle technical challenges, such as location estimation, minimizing power loss, and efficient frequency allocation.

### 3.1. Threat Model

In the context of the *LazyFrog* mechanism within a remote charging service environment, we delineated three principal threats under the assumption that attackers situated within the coverage area of a charging station are capable of intercepting all frequencies utilized by the station.

The first threat, as illustrated in Figure 3A, pertains to the potential for attackers to freely manipulate the frequency range of the charging station. As illustrated in Figure 3, if the power leakage is detected at frequency f1 with subsequent hopping to f3, and an attacker then shifts to f3, hopping re-occurs. This tactic enables attackers to exploit the proposed mechanism, thereby destabilizing the system.

The second threat, as illustrated in Figure 3B, involves active attackers monopolizing a particular frequency. By continuously drawing power intended for *LazyFrog* from a charging station at a designated frequency, these attackers impede the allocation or frequency hopping for new devices.

The third threat, as presented in Figure 3C, highlights the possibility of extreme attackers targeting all frequencies utilized by the charging station. Such attackers can significantly degrade the quality of service, or in severe cases, paralyze it through continuous frequency alterations, such as in a denial-of-service (DoS) attack. This results in constant hopping or prevents the remote charging allocation to new devices. In scenarios wherein these threats materialize, the charging efficiency of the APs severely declines, thus prompting the service provider to guide users out of the impacted charging station coverage area and allocate them to a new station to continue service.

During the process of developing the threat model, we assumed that attackers cannot physically access the servers, transmitters, or receivers of the wireless charging system. However, attackers are capable of intercepting the charging frequencies used by the system within the coverage area. This foundational assumption informed the development of the threat model, which specifically addresses vulnerabilities inherent to the remote charging environment and operational continuity of the *LazyFrog* mechanism.

### 3.2. Architecture for *LazyFrog* Mechanism

The core aspects of the *LazyFrog* mechanism are illustrated in Figure 4. The architecture of the *LazyFrog* mechanism consists of three main components: devices, charging stations (APs), and the *LazyFrog* server. Each component plays a crucial role in the overall functioning of the system, and their interactions are carefully orchestrated to ensure secure and efficient wireless power transmission.

Devices: The devices in the *LazyFrog* mechanism are the end-users or the recipients of wireless power. These can be smartphones, wearables, or any other electronic devices capable of receiving wireless power. Each device is equipped with a wireless charging receiver and a Wi-Fi modem for communication with the charging stations and the *LazyFrog* server.*LazyFrog* Server: The *LazyFrog* server is the central component of the *LazyFrog* mechanism, responsible for managing the entire wireless power transmission mechanism. The server maintains a database of registered devices, charging stations, and their associated information. It authenticates devices and users to prevent unauthorized access to the system. Users authenticate themselves with their identification (ID) through a login form, and upon authentication, the server requests the RSSI values for the Wi-Fi SSID from the user device. The server receives location information from the devices in the form of Wi-Fi RSSI values and runs the relative-distance estimation algorithm to determine the most suitable charging station for each device. Subsequently, the server evaluates the frequency interference by comparing the expected charging power with the actual charging power. If interference is detected, it dynamically assigns frequencies to the devices and charging stations and initiates frequency hopping to mitigate power theft.Charging Stations (APs):The charging stations, also referred to as access points (APs), are responsible for transmitting wireless power to the devices. They are equipped with wireless charging transmitters and Wi-Fi modems for communication with the devices and the *LazyFrog* server. The charging stations receive dynamic frequency assignments from the server based on the current threat and power loss detection. The charging stations transmit power to the devices using the assigned frequencies and follow the instructions provided by the server for frequency hopping when necessary.

### 3.3. Implementation

The *LazyFrog* mechanism is implemented as a comprehensive wireless power transmission mechanism that prioritizes security and efficiency, while acknowledging the practical limitations of wireless charging technology. The system consists of three key components as illustrated in Figure 5: the patrol mechanism, the AP allocation mechanism, and the detection and hopping mechanism. It is important to note that the actual efficiency of wireless power transmission cannot reach 100% due to various factors such as signal attenuation, environmental interference, and inherent energy losses during the transmission process. The *LazyFrog* mechanism is designed to optimize the efficiency of wireless charging within these realistic constraints. The patrol mechanism was implemented to enable charging stations to communicate with each other and identify safe frequencies for power transmission. This process involves periodic power transmission tests among the charging stations, which helps to detect and isolate compromised frequencies. The server maintains a list of safe frequencies based on the information provided by the charging stations, ensuring a secure network for wireless power transmission while minimizing energy losses due to interference or unauthorized access. The AP allocation mechanism was developed to optimize the connection between the device and the nearest charging station, thereby maximizing the efficiency of power delivery. When a device requests wireless power, it sends its location data to the server using RSSI values. The server analyzes the location data and assigns the device to the most suitable AP based on the RF distance and device characteristics. If the device moves during the charging process, the server dynamically reassigns it to a new AP that provides a better RSSI value, ensuring a stable and efficient charging experience by reducing unnecessary energy losses due to signal attenuation or suboptimal connections. The detection and hopping mechanism was implemented to actively monitor the power received by the device and compare it with the expected power output calculated by the server, taking into account the practical limitations of wireless power transmission. If a significant discrepancy is detected, indicating potential unauthorized access or power loss beyond the expected levels, the server promptly changes the transmission frequency to a safe one from the pre-established list.

#### 3.3.1. Patrol Mechanism

The patrol mechanism is a cornerstone of the *LazyFrog* system, playing a critical role in safeguarding the availability and security of the wireless charging infrastructure. It systematically conducts tests on power transmissions between charging stations to identify the presence of unauthorized users. In this process, each charging station monitors the power it transmits to others. The expected power transmitted between charging stations is calculated using the Friis transmission equation:(1)$$P_{r}=P_{t}\cdot G_{t}\cdot G_{r}\cdot {\left(\frac{\lambda }{4\pi d}\right)}^{2}$$
where *P_r_* is the received power, *P_t_* is the transmitted power, *G_t_* and *G_r_* are the gains of the transmitting and receiving antennas, respectively, λ is the wavelength of the signal, and *d* is the distance between the transmitting and receiving stations.

As shown in Figure 6, If the transmitted power falls below the expected level, it is highly probable that an unauthorized user is occupying that frequency band. Detecting such a discrepancy leads to the frequency being labeled as abnormal and reported to the server, which then restricts access to this compromised frequency. Charging stations routinely carry out the patrol procedure, thus refining the list of safe frequencies stored on the server. Incorporating the patrol mechanism provides a pivotal advantage in navigating around attackers who may be monopolizing specific frequencies. This strategic utilization of the patrol mechanism ensures that the *LazyFrog* mechanism adeptly circumvents attackers, thereby maintaining a secure and efficient charging environment across the network.

#### 3.3.2. AP Allocation

The AP allocation mechanism is implemented to optimize the connection between the device and the nearest charging station. As shown in Figure 7, When a device requests wireless power, it sends its location data to the *LazyFrog* server using RSSI (Received Signal Strength Indicator) values obtained from nearby APs. The server receives the RSSI values and runs the relative-distance estimation algorithm to calculate the distances between the device and each AP. The RSSI values are converted to distances using the log-distance path loss model:(2)$$\mathrm{RSSI}=-10n\log_{10}\left(d\right)+A$$

The server uses the calculated distances to estimate the relative-position of the device using the trilateration technique. Trilateration determines the device’s position by solving a system of equations based on the distances from at least three known AP locations. The equations are given by
(3)$$\begin{array}{c} {\left(x-x_{i}\right)}^{2}+{\left(y-y_{i}\right)}^{2}=d_{i}^{2},\;\mathrm{for}\;i=1,2,\ldots ,n \end{array}$$
where (x,y) is the estimated position of the device, $\left(x_{i},y_{i}\right)$ are the coordinates of the *i*-th AP, and $d_{i}$ is the calculated distance between the device and the *i*-th AP. The algorithm then determines the AP that is closest to the device, taking into account factors such as signal strength, interference, and obstructions.

Once the server identifies the most suitable AP for the device, it assigns the device to that AP and establishes a connection between them. The server also selects a safe frequency from the list maintained by the patrol mechanism and informs both the device and the AP about the assigned frequency for power transmission.

During the charging process, the server continuously monitors the RSSI values reported by the device. If the device moves and the RSSI values indicate that another AP is closer or provides a better signal strength, the server dynamically reassigns the device to the new AP. This ensures that the device always receives power from the most suitable AP, maintaining a stable and efficient charging experience.

#### 3.3.3. Detecting and Hopping

The detection and hopping mechanism is a critical component of the *LazyFrog* system that actively monitors the power received by the devices and compares it with the expected power calculated by the server. This mechanism is designed to detect any discrepancies or power losses that may indicate unauthorized access or interference in the wireless power transmission process. The expected power received by a device is calculated using the Friis transmission equation:(4)$$P_{r}=P_{t}\cdot G_{t}\cdot G_{r}\cdot {\left(\frac{\lambda }{4\pi d}\right)}^{2}$$
where $P_{r}$ is the received power, $P_{t}$ is the transmitted power, $G_{t}$ and $G_{r}$ are the gains of the transmitting and receiving antennas, respectively, λ is the wavelength of the signal, and *d* is the distance between the charging station and receiver. The server periodically receives power measurements from the device, including the actual power received and the round-trip time (RTT) data. The server uses this information to calculate the expected power output based on the established power transmission rate and the designated check frequency interval. If the actual power received by the device significantly differs from the expected power output (e.g., a difference of more than 50%), then the server identifies this as a potential power loss or unauthorized access attempt. In such cases, the server triggers the hopping mechanism to change the transmission frequency and prevent further power theft. The hopping mechanism is implemented using a frequency change algorithm that selects a new safe frequency from the list maintained by the patrol mechanism. The server sends the new frequency information to both the device and the AP, instructing them to switch to the new frequency for power transmission.

The detection and hopping mechanism also takes into account the power losses caused by transmission delays. The server uses the RTT data received from the device to calculate the power loss attributed to such delays. This calculated power loss is then factored into the expected power output calculation, ensuring a more accurate estimation of the power that should be received by the device.

#### 3.3.4. Signal Attenuation Monitoring and Real-time Position Analysis

The *LazyFrog* mechanism employs a combination of real-time position estimation and signal attenuation monitoring to ensure the security and reliability of the wireless charging system. The server performs trilateration to estimate the position of the receiver based on the RSSI values reported by the receiver, while simultaneously monitoring the actual power received by the receiver and comparing it with the expected power calculated using the Friis transmission equation. The position estimation is performed using the trilateration technique described in the AP allocation section Equations (2) and (3).

For signal attenuation monitoring, the server uses the Friis transmission equation Equation (Equation 1) to calculate the expected power received by the receiver based on the estimated position and the transmitted power. The server compares the actual power received by the receiver ($P_{actual}$) with the expected power calculated using the Friis transmission equation ($P_{expected}$). If the difference between $P_{actual}$ and $P_{expected}$ exceeds a predefined threshold ($P_{threshold}$), it indicates significant signal attenuation.

The server uses the following algorithm to detect signal attenuation and potential attack situations:Estimate the receiver’s position using trilateration based on the RSSI values reported by the receiver Equations (2) and (3).Calculate the expected power received by the receiver ($P_{expected}$) using the Friis transmission equation Equation (Equation 1) based on the estimated position and the transmitted power.Compare the actual power received by the receiver ($P_{actual}$) with $P_{expected}$.If the difference between $P_{actual}$ and $P_{expected}$ exceeds $P_{threshold}$, it indicates significant signal attenuation.Calculate the distance between the current estimated position and the previous estimated position ($position_{diff}$).If $position_{diff}$ exceeds a predefined position threshold ($position_{threshold}$) without a corresponding change in the signal attenuation, it indicates a potential attack situation.

If signal attenuation caused by a bag or pocket is determined, the server can send a notification to the receiver requesting the removal of the obstacle. The receiver can then take appropriate actions, such as repositioning itself or removing the obstacles, to improve the signal quality and ensure efficient wireless charging.

If a potential attack situation is detected, the server initiates defensive measures, such as frequency hopping, to protect the wireless charging system. It switches to a secure channel and notifies the user of the potential threat. The server may also temporarily suspend the wireless charging process until the threat is resolved.

## 4. Evaluation

This section provides a thorough evaluation of the efficiency of the *LazyFrog* mechanism using rigorous experimentation and analysis through software simulations. This approach was chosen to incorporate various parameters related to the *LazyFrog* mechanism and attacker scenarios, enabling the generation of reliable evaluation results and improving the efficiency of the evaluation experiments. The research hypothesis is that a system incorporating the *LazyFrog* mechanism significantly reduces power loss due to unauthorized access and improves power transmission efficiency. For verification, we investigated the power loss associated with frequent frequency changes, power loss during the synchronization process, and the impact of unauthorized users.

In particular, we evaluated the efficiency and security of the *LazyFrog* mechanism and compared it with the traditional wireless power transmission mechanism in a simulated environment. The simulation setup allowed us to test the performance of both mechanisms under various conditions, such as different frequency ranges, power transmission rates, and attacker capabilities. By using simulations, we were able to cover a wide range of scenarios and obtain comprehensive results that might be difficult to achieve through physical experiments alone.

### 4.1. Evaluation Overview

This evaluation process involved rigorous experimentation and analysis, with a focus on randomizing experimental conditions and initial settings to minimize selection bias and ensure the reliability and validity of the results.

The simulation environment was designed to closely mimic real-world scenarios, incorporating multiple charging stations (APs), a single receiver (device), and an optional attacker. The number of charging stations, simulation size, and other relevant parameters were customizable, enabling the evaluation of the systems’ performance under various conditions. The operational framework of the charging mechanism was defined by several key parameters, including the frequency change interval, synchronization period, power output, and simulation duration. These parameters were carefully selected based on the characteristics of the *LazyFrog* mechanism and the traditional wireless power transmission mechanism, as well as the requirements of the simulation environment.

External variables that could potentially influence system performance, such as distance within the virtual space, power transmission and reception intensity, and number of devices, were meticulously monitored and adjusted throughout the experiment. These variables were considered as covariates in the analysis of results and were incorporated into the statistical models to isolate their effects and obtain a more accurate evaluation of the *LazyFrog* mechanism’s performance.

The efficiency of the system was thoroughly examined, focusing on the total power transmitted, which represented the amount of power successfully received by end-users throughout the simulation, and the total power loss, primarily due to synchronization downtime. In addition to efficiency, the security aspect of the *LazyFrog* mechanism was rigorously investigated. The number of successful attacks, the amount of power stolen by unauthorized users, and the detection rate of the *LazyFrog* mechanism were measured and analyzed.

To ensure the statistical significance and reliability of the results, each experiment was repeated under various environmental conditions and attack scenarios. The results obtained from these repetitions were averaged to derive representative indicators and were subjected to tests for normality and homogeneity of variances to confirm their suitability for statistical analysis.

### 4.2. Experimental Environment Setup

The software simulation was developed using Python version 3.9 on a Linux-based system. Python was chosen for its extensive library support, particularly for scientific computing and data analysis, which are crucial for the simulation and evaluation of the *LazyFrog* mechanism. Linux was selected as the operating system due to its stability, security, and compatibility with various hardware configurations, ensuring a reliable and consistent environment for the experiments. To ensure consistency during the simulation development and execution process, all system clocks were precisely synchronized using the same NTP server.

#### 4.2.1. Traditional Wireless Power Transmission Mechanism

The traditional wireless power transmission mechanism primarily utilizes the coupled magnetic resonance (CMR) approach, thus efficiently transmitting power between the charging station and receiver through the matching of resonant frequencies. This mechanism has a basic configuration that operates at variable frequencies, and a mechanism that enhances security by applying a frequency hopping mechanism through a pseudo-random code sequence was considered. In this evaluation, the traditional wireless power transmission mechanism simulation was designed to closely mimic real-world scenarios. The traditional mechanism employed a fixed frequency change period and an interval for checking the status of frequency changes to maintain synchronization between the charging stations and the receiver. The simulation environment allowed for customizable settings such as the number of charging stations, simulation size, and various other parameters to evaluate the system’s performance under different conditions. The traditional mechanism’s performance was evaluated based on metrics such as total transmitted power, receiver received power, attacker stolen power, number of attacks, total frequency change count, and power loss due to delays.

The traditional wireless power transmission mechanism simulation consists of several key components:Server: The server component manages the overall system, including charging station and receiver connections, frequency assignments, and monitoring power transmission. It maintains a database of connected devices and their associated information.charging stations (APs): Multiple charging stations are deployed in the simulation environment, each capable of transmitting power at a specific frequency. The charging stations follow the frequency hopping sequence provided by the server and transmit power to the receiver when the frequencies match.Receiver(Devices): The receiver is a single device that receives power from the charging stations when the frequency is matched. It communicates with the server to report the received power and any detected power loss.Attacker (optional): When enabled, the attacker attempts to intercept and steal power from the wireless power transmission mechanism by matching the transmission frequency.

#### 4.2.2. *LazyFrog* Wireless Power Transmission Mechanism

The *LazyFrog* simulation environment was designed to closely resemble the traditional system, with the addition of the *LazyFrog* server component. The server manages the connections between devices and charging stations, measures power consumption to prevent unauthorized use, and dynamically adjusts the wireless power transmission frequency when necessary.

The *LazyFrog* wireless power transmission mechanism simulation consists of the following key components:*LazyFrog* Server: The *LazyFrog* server is an enhanced version of the server component in the traditional system. In addition to managing device connections and frequency assignments, it actively monitors power consumption and adapts the frequency hopping sequence based on detected threats or power loss. The server maintains a list of safe frequencies and assigns them to the charging stations and receivers dynamically.charging stations (APs): Similar to the traditional system, multiple charging stations are deployed in the simulation environment. However, in the *LazyFrog* system, the charging stations communicate with the server to receive dynamic frequency assignments based on the current threat and power loss detection.*LazyFrog* Receiver(Devices): The *LazyFrog* receiver is an upgraded version of the receiver in the traditional system. It actively monitors the received power and reports any discrepancies or power loss to the server in real-time. The receiver also adapts its receiving frequency based on the server’s instructions to maintain a secure and efficient power transmission.Attacker (optional): When enabled, the attacker attempts to intercept and steal power from the wireless power transmission mechanism by matching the transmission frequency, similar to the traditional system.

#### 4.2.3. Setting Variables

This study aimed to provide a comprehensive evaluation of the *LazyFrog* mechanism’s performance and its advantages over traditional wireless power transmission mechanisms under diverse conditions. In both the traditional and *LazyFrog* simulations, various external variables that could influence system performance were identified in the initial stages of the experimental design.

These variables were considered as covariates in the analysis of results to adjust for their effects, allowing for a clearer evaluation of the pure effect of the *LazyFrog* mechanism. Table 1 below summarizes the variables and their respective ranges used in the simulations.

Frequency (FREQUENCY_RANGE): The simulations used frequencies within the range of 5–30 kHz. The frequency range variable was varied from 5 to 30 kHz in increments of 5 kHz to evaluate the impact of different frequency ranges on the system’s performance.Frequency change period (FREQUENCY_CHANGE_PERIOD): The frequency change period represents the time interval at which the charging stations change their operating frequency in the traditional system. The traditional simulations considered frequency change periods ranging from 0.1 to 5 s in increments of 0.1 s to evaluate the effect of different frequency change intervals on the system’s efficiency and security.Attack scan intervals (ATTACK_SCAN_INTERVALS): The attack scan intervals represent the time interval between each scan performed by the attacker in an attempt to intercept and steal power from the wireless power transmission mechanism. The simulations considered attack scan intervals ranging from 0.007 to 0.03 s in increments of 0.001 s to evaluate the system’s security against attackers with varying scanning capabilities.Power transmission rate (POWER_TRANSMISSION_RATE): The power transmission rate defines the rate at which power is transmitted by the charging stations in the wireless power transmission mechanism. The simulations were conducted with different power transmission rates, ranging from 5 to 15 W in increments of 2 W, to evaluate the systems’ efficiency and performance under varying power demands.Simulation size (SIMULATION_SIZE): The simulation size represents the dimensions of the virtual space in which the simulation is conducted. The simulations considered varying simulation sizes, such as (10,10), (20,20), (30,30), (40,40), and (50,50) units, to evaluate the system’s performance and scalability in different deployment scenarios.

### 4.3. Efficiency Measurement and Security Evaluation

This section details the experiments conducted to evaluate the efficiency and security of the *LazyFrog* mechanism. Both the traditional wireless power transmission mechanism and the *LazyFrog* power transmission mechanism were configured and operated within the same settings. The focus of this experiment was to quantify the ratio of power loss caused by attackers.

It is important to note that 100% efficiency in wireless power transmission is fundamentally unachievable. In our experiments, we utilized the 80% efficiency value measured in previous studies as a baseline for conducting the evaluation simulations [2].

#### 4.3.1. Simulation Experiments for Efficiency and Security Evaluation of the Traditional Mechanism

To evaluate the efficiency and security of the traditional wireless power transmission mechanism, a series of simulation experiments were conducted. The simulations were designed to measure the power transmission efficiency and evaluate the system’s security to attacks under various operating conditions. The key parameter varied in the simulation experiments was the frequency change period, which represents the time interval between frequency changes in the system.

During the simulations, the power transmission efficiency was measured by calculating the ratio of the total power received by the receiver to the total power transmitted by the charging station. The security aspect was evaluated by introducing an attacker into the system and measuring the amount of power stolen by the attacker.

The simulation results revealed that the traditional wireless power transmission mechanism’s efficiency and security are significantly influenced by the frequency change period. As the frequency change period increased, the power transmission efficiency improved, but the system became more vulnerable to attacks.

In the scenario with the shortest frequency change period of 0.1 s, as illustrated in Figure 8, the power loss due to synchronization delay was high, reaching 1.07% of the received power. However, the attacker’s stolen power was relatively low at 1.063% of the received power.

In the scenario with the longest frequency change period of 5.0 s, the power loss due to synchronization delay was reduced to 0.029% of the received power, indicating a significant improvement in power transmission efficiency. However, the attacker’s stolen power reached a staggering 67.284% of the received power, exposing the severe security vulnerabilities of the traditional mechanism.

These results demonstrate that the traditional wireless power transmission mechanism faces a trade-off between efficiency and security. Shorter frequency change periods provide better security against attacks but result in higher power losses due to synchronization delays. Conversely, longer frequency change periods improve power transmission efficiency but make the system more vulnerable to attacks.

#### 4.3.2. Simulation Experiments for Efficiency and Security Evaluation of the *LazyFrog* Mechanism

To evaluate the efficiency and security of the *LazyFrog* mechanism, a series of simulation experiments were conducted. The *LazyFrog* mechanism enhances the capabilities of the standard wireless power transmission mechanism by adopting a selective frequency change strategy. In these experiments, frequency changes occurred only when the power loss exceeded a predetermined threshold or when an attack was detected.

During the simulations, key metrics such as the frequency of changed frequencies, the number of attacks, and the power loss ratio were collected. The simulation results demonstrated that the *LazyFrog* mechanism operates at high efficiency, achieving minimal power loss due to synchronization delays.

The best performance, as illustrated in Figure 9, was observed with an attack scan interval of 0.028 s, resulting in a power transmission of 114.113 W, an average of 5.8 attacks, and an average total power loss due to delay of 0.023 W (0.02% of received power).

However, the worst-case scenario in terms of both efficiency and security was encountered with the shortest attack scan interval of 0.007 s. In this scenario, the attacker’s frequent scanning induced a high number of synchronizations, resulting in an average total power loss due to delay of 0.106 W (0.093% of received power) and an average of 26.4 successful attacks, with the attacker stealing an average of 0.05 W of power. This suggests that shorter scan intervals employed by the attacker can simultaneously degrade the efficiency and security of the *LazyFrog* mechanism.

#### 4.3.3. Analysis of Efficiency Evaluation Results

The results of the one-way analysis of variance conducted to analyze the difference in power transmission efficiency between groups according to the frequency change period demonstrated that the main effect of the frequency change period was statistically significant (F(49,450)=11.48, p<0.001, $\eta^{2}=0.56$), as shown in Table 2. Post-hoc analysis demonstrated that the group with the *LazyFrog* mechanism applied exhibited significantly higher power transmission efficiency under all conditions of frequency change periods when compared with the traditional mechanism group (p<0.001). Particularly, with a decrease in the length of the frequency change period, the efficiency difference between the two groups increased, thus indicating that the adaptive frequency change of the *LazyFrog* mechanism technique contributes to improving system efficiency.

The efficiency evaluation results demonstrate that the *LazyFrog* mechanism outperforms the traditional wireless power transmission mechanism in terms of power transmission efficiency. The adaptive frequency hopping strategy employed by *LazyFrog* allows it to minimize power loss due to synchronization delays and maintain a higher power transmission rate compared to the traditional mechanism.

The impact of various parameters on the efficiency of both mechanisms was analyzed. For the traditional mechanism, the frequency change period had a significant effect on the power transmission efficiency, with shorter periods resulting in higher power loss due to synchronization delays. The frequency range, attack scan intervals, and power transmission rate had minimal impact on the efficiency of the traditional mechanism.

In the case of the *LazyFrog* mechanism, the attack scan interval played a crucial role in determining the power transmission efficiency. Shorter scan intervals employed by the attacker resulted in lower efficiency for *LazyFrog* due to the increased number of frequency changes required. The frequency range, power transmission rate, and simulation size had a relatively minor impact on *LazyFrog*’s efficiency compared to the attack scan interval.

Overall, the efficiency evaluation results highlight the superiority of the *LazyFrog* mechanism in terms of power transmission efficiency, particularly in scenarios with longer attack scan intervals and fewer frequency changes. However, it is important to note that shorter attack scan intervals employed by the attacker can simultaneously degrade the efficiency and security of the *LazyFrog* mechanism, as demonstrated in the worst-case scenario with an attack scan interval of 0.007 s.

To analyze the efficiency evaluation results, a one-way analysis of variance (ANOVA) was conducted to compare the power transmission efficiency between the traditional wireless power transmission mechanism and the *LazyFrog* mechanism under various conditions. The results of the ANOVA are presented in Table 2. The power transmission efficiency (η) is calculated using the following formula:(5)$$\eta =\left(\frac{P_{\mathrm{received}}}{P_{\mathrm{transmitted}}}\right)\times 100$$
where $P_{\mathrm{received}}$ is the power received by the receiver and $P_{\mathrm{transmitted}}$ is the total power transmitted by the charging station.

The ANOVA model can be expressed as follows:(6)$$Y_{ij}=\mu +\alpha_{i}+\varepsilon_{ij}$$
where $Y_{ij}$ is the power transmission efficiency of the *j*th observation in the *i*th group (traditional or *LazyFrog*), μ is the overall mean efficiency, $\alpha_{i}$ is the effect of the *i*th group, and $\varepsilon_{ij}$ is the random error term.

The ANOVA results demonstrated a statistically significant difference in power transmission efficiency between the two mechanisms (F(1,98)=2076.28, p<0.001, $\eta^{2}=0.95$). The *F*-statistic is calculated as the ratio of the between-group variance to the within-group variance, and the *p*-value indicates the probability of observing such an extreme *F*-statistic under the null hypothesis of no difference between the groups. The effect size, $\eta^{2}$, represents the proportion of variance in the dependent variable (efficiency) that is explained by the independent variable (mechanism type).

The *LazyFrog* mechanism (M=75.97%, SD=1.39%) demonstrated significantly higher power transmission efficiency compared to the traditional mechanism (M=69.63%, SD=18.77%). The mean (*M*) and standard deviation (SD) provide a summary of the efficiency values for each mechanism.

Further analysis was conducted to investigate the impact of different parameters on the efficiency of each mechanism.

For the traditional mechanism, a one-way ANOVA showed a significant effect of the frequency change period on power transmission efficiency (F(49,450)=283.87, p<0.001, $\eta^{2}=0.97$), as shown in Table 3. The frequency change period ($T_{\mathrm{fc}}$) is the time interval between consecutive frequency changes in the traditional mechanism. Post-hoc comparisons using Tukey’s HSD test indicated that shorter frequency change periods led to significantly lower efficiency compared to longer periods (p<0.05). Tukey’s HSD test controls the family-wise error rate when making multiple pairwise comparisons between group means.

For the *LazyFrog* mechanism, a one-way ANOVA demonstrated a significant effect of the attack scan interval ($T_{\mathrm{scan}}$) on power transmission efficiency (F(23,216)=214.09, p<0.001, $\eta^{2}=0.96$). The attack scan interval is the time interval between consecutive scans performed by the attacker to intercept and steal power. Shorter attack scan intervals led to higher power loss and lower efficiency compared to longer intervals (p<0.05). This indicates that the efficiency of the *LazyFrog* mechanism is sensitive to changes in the attack scan interval parameter, with shorter attack scan intervals employed by the attacker resulting in decreased efficiency.

The frequency range ($F_{\mathrm{range}}$), power transmission rate ($P_{\mathrm{rate}}$), and simulation size ($S_{\mathrm{size}}$) did not have a statistically significant impact on the efficiency of either mechanism (p>0.05), suggesting that these parameters have minimal influence on the power transmission efficiency in the given experimental setup.

#### 4.3.4. Analysis of the Security Evaluation Results

The security evaluation results demonstrate that the *LazyFrog* mechanism provides enhanced protection against unauthorized access and power theft compared to the traditional wireless power transmission mechanism.

In the traditional mechanism, the attacker success rate was significantly higher, with the attacker managing to steal up to 67.284% of the received power in the worst-case scenario. The number of successful attacks was also higher in the traditional mechanism, reaching up to 18.8 attacks with a frequency range of 5 kHz.

On the other hand, the *LazyFrog* mechanism limited the attacker’s success rate and the amount of power stolen. The highest amount of power stolen by the attacker in the *LazyFrog* mechanism was 0.054% of the received power, which is considerably lower than the traditional mechanism. The number of successful attacks was also reduced, with a maximum of 28.6 attacks observed with a simulation size of (50, 50).

To evaluate the security posed by unauthorized access and power theft, an independent samples *t*-test was conducted to compare the attacker stolen power and the number of successful attacks between the traditional wireless power transmission mechanism and the *LazyFrog* mechanism.

The *t*-test model can be expressed as follows:(7)$$t=\frac{M_{1}-M_{2}}{s_{p}\times \sqrt{\frac{1}{n_{1}}+\frac{1}{n_{2}}}}$$
where $M_{1}$ and $M_{2}$ are the means of the two groups, $s_{p}$ is the pooled standard deviation, $n_{1}$ and $n_{2}$ are the sample sizes of the two groups, and √ denotes the square root function.

The *t*-test results indicated a statistically significant difference in the attacker stolen power ($P_{\mathrm{stolen}}$) between the two mechanisms (t(98)=28.12, p<0.001, Cohen’s d=5.62). The traditional mechanism (M=11.31%, SD=9.02%) experienced significantly higher power theft compared to the *LazyFrog* mechanism (M=0.05%, SD=0.01%). Cohen’s *d* is a measure of effect size, representing the standardized difference between the means of the two groups.

Similarly, there was a significant difference in the number of successful attacks ($N_{\mathrm{attacks}}$) between the two mechanisms (t(98)=12.37, p<0.001, Cohen’s d=2.47). The traditional mechanism (M=14.08, SD=1.97) faced a significantly higher number of successful attacks compared to the *LazyFrog* mechanism (M=26.83, SD=0.91).

Further analysis using one-way ANOVAs demonstrated that the frequency change period ($T_{\mathrm{fc}}$) had a significant effect on the attacker stolen power (F(49,450)=285.27, p<0.001, $\eta^{2}=0.97$) and the number of successful attacks (F(49,450)=5.01, p<0.001, $\eta^{2}=0.35$) in the traditional mechanism, as indicated in Table 4. Longer frequency change periods were associated with higher power theft and more successful attacks (p<0.05), as the attacker had more opportunities to intercept and steal power during the infrequent frequency changes.

For the *LazyFrog* mechanism, the attack scan interval ($T_{\mathrm{scan}}$) had a significant impact on the attacker stolen power (F(23,216)=214.09, p<0.001, $\eta^{2}=0.96$) and the number of successful attacks (F(23,216)=214.09, p<0.001, $\eta^{2}=0.96$), as presented in Table 4. Shorter attack scan intervals led to higher power theft and more successful attacks compared to longer intervals (p<0.05). This is because shorter attack scan intervals allow the attacker to more frequently search for and exploit vulnerabilities in the wireless power transmission mechanism.

### 4.4. Sensitivity Analysis

In this study, a comprehensive sensitivity analysis was conducted to evaluate the robustness and performance of the *LazyFrog* mechanism under various experimental conditions. As shown in Table 5, the analysis involved testing the mechanism’s performance by varying key parameters such as the frequency change period, attack scan intervals, power transmission rate, and simulation size.

The sensitivity analysis was performed using both the *LazyFrog* mechanism and the traditional wireless power transmission mechanism for comparison. The performance metrics evaluated in the sensitivity analysis included:Average total transmitted powerAverage receiver received powerAverage attacker stolen powerAverage number of attacksAverage total frequency change countAverage total power loss due to delayAverage percentage of power lost due to delayAverage percentage of power stolen by attacker

The results of the sensitivity analysis demonstrated that the *LazyFrog* mechanism maintained stable performance across a wide range of experimental conditions. When the frequency change period was varied from 0.1 to 5 s, the average receiver received power of the traditional mechanism decreased significantly from 111.764 W to 68.334 W, while the average attacker stolen power increased from 1.187 W to 45.793 W. In contrast, the *LazyFrog* mechanism maintained a stable average receiver received power above 113.98 W and an average attacker stolen power below 0.1 W, as it only changes the frequency when a threat is detected.

Similarly, when the attack scan intervals were increased from 0.007 to 0.03 s, the *LazyFrog* mechanism maintained a stable average receiver received power ranging from 113.988 W to 114.111 W, while the traditional mechanism showed a larger variation from 107.472 W to 111.457 W. The average attacker stolen power for the *LazyFrog* mechanism remained very low, between 0.01 W and 0.05 W, whereas the traditional mechanism had a higher range from 1.631 W to 6.439 W. This demonstrates that the *LazyFrog* mechanism can provide stable and secure power transmission regardless of the attack scan intervals.

The sensitivity analysis also demonstrated that both the *LazyFrog* mechanism and the traditional mechanism exhibited a proportional increase in average receiver received power as the power transmission rate increased. However, the traditional mechanism’s average attacker stolen power increased significantly from 5.907 W to 21.024 W, while the *LazyFrog* mechanism maintained a very low level, ranging from 0.05 W to 0.164 W. This indicates that the *LazyFrog* mechanism can maintain stable security regardless of the power transmission rate.

Based on the sensitivity analysis results, it can be inferred that the performance of the *LazyFrog* mechanism is robust and consistent across a wide range of experimental conditions and confounding variables, providing superior performance and security compared to the traditional wireless power transmission mechanism.

### 4.5. Interpretation and Discussion of Results

The experimental results of this study demonstrated that the *LazyFrog* mechanism is a promising solution capable of simultaneously enhancing the security and efficiency of wireless power transmission mechanisms.

The one-way ANOVA results revealed that the *LazyFrog* mechanism achieved significantly higher power transmission efficiency compared to the traditional wireless power transmission mechanism (F(1,98)=2076.28, p<0.001, $\eta^{2}=0.95$). This finding highlights the effectiveness of the adaptive frequency hopping strategy employed by *LazyFrog* in optimizing power delivery and reducing losses due to synchronization delays.

Moreover, the independent samples *t*-test results indicated that the *LazyFrog* mechanism provided superior protection against unauthorized access and power theft compared to the traditional mechanism. The *LazyFrog* mechanism experienced significantly lower power theft (t(98)=28.12, p<0.001, Cohen’s d=5.62) and fewer successful attacks (t(98)=12.37, p<0.001, Cohen’s d=2.47) than the traditional mechanism. These results suggest that the adaptive frequency hopping and real-time power monitoring features of *LazyFrog* are highly effective in detecting and mitigating security threats in wireless power transmission mechanisms.

The sensitivity analysis results further reinforced the robustness and versatility of the *LazyFrog* mechanism. The mechanism maintained stable average receiver received power across a wide range of experimental conditions, including variations in frequency range (113.980 W to 113.993 W), attack scan intervals (113.988 W to 114.113 W), power transmission rate (113.988 W to 341.933 W), and simulation size (113.979 W to 113.998 W).

This adaptability is crucial for the practical implementation of the *LazyFrog* mechanism in real-world wireless power transmission scenarios, where environmental factors and user behaviors may vary significantly.

However, it is important to acknowledge the limitations of this study. The experiments were conducted in a controlled simulation environment, which may not fully capture the complexities and uncertainties of real-world wireless power transmission scenarios.

Future research should focus on validating the *LazyFrog* mechanism’s performance in practical settings, considering factors such as device heterogeneity, user mobility, and environmental interference.

Furthermore, the current study primarily focused on the efficiency and security aspects of the *LazyFrog* mechanism. Additional research is needed to investigate other important factors, such as scalability, compatibility with existing wireless power transmission standards, and user experience.

## 5. Conclusions

This study was focused on the stability and efficiency of wireless power transmission technologies to address the major challenges in this field. The increase in electronic devices and electricity-powered mobility solutions in modern society highlights the importance of wireless power transmission infrastructure, thus increasing the consideration of reliability and safety for users. The proposed *LazyFrog* mechanism can overcome the existing issues of wireless power transmission technology by extending its application scope and efficient charging system design. The *LazyFrog* mechanism developed in this research identifies threats that may arise during the wireless power transmission process and effectively blocks unauthorized access by adjusting the timing of frequency changes. These features are critical for enhancing the availability of wireless power transmission technologies and reducing technical barriers. The experimental results demonstrate the effectiveness of the *LazyFrog* mechanism, which outperforms existing wireless power transmission technologies. This enhances the practicality and reliability of wireless power transmission technology and is expected to contribute to improving its widespread adoption and integration into society as well as user experience.

## Figures and Tables

**Figure 1 sensors-24-02571-f001:**
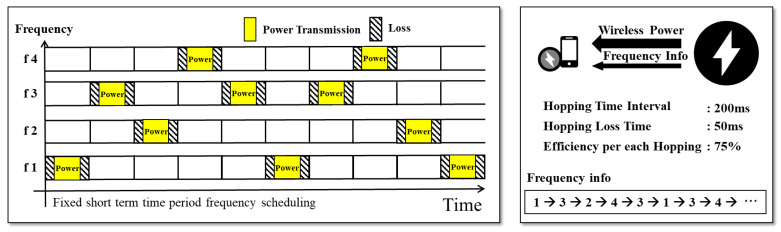
Synchronization delay of wireless charging system based on frequency scheduling.

**Figure 2 sensors-24-02571-f002:**
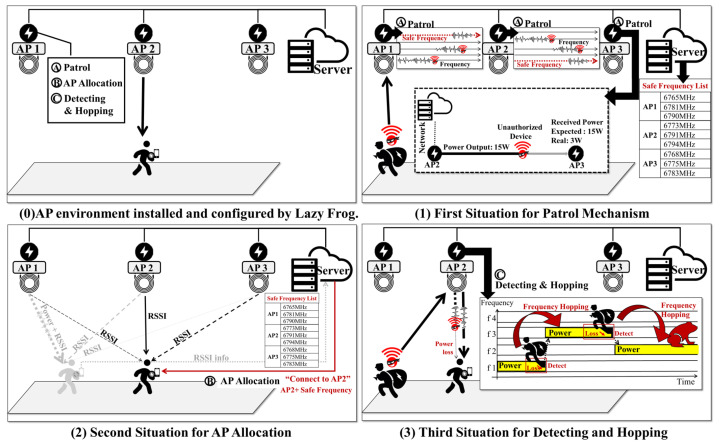
Overall wireless charging schemes based on *LazyFrog*.

**Figure 3 sensors-24-02571-f003:**
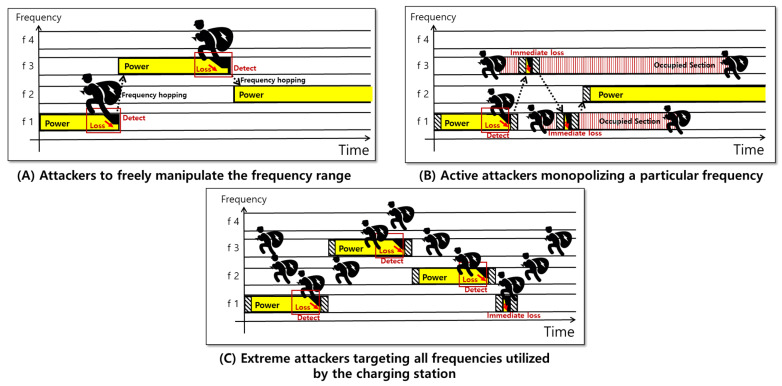
Three primary threat models to the wireless charging system.

**Figure 4 sensors-24-02571-f004:**
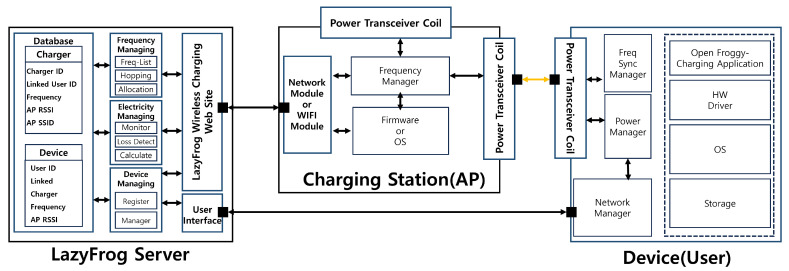
Overall architecture of the wireless charging system for the *LazyFrog* mechanism.

**Figure 5 sensors-24-02571-f005:**
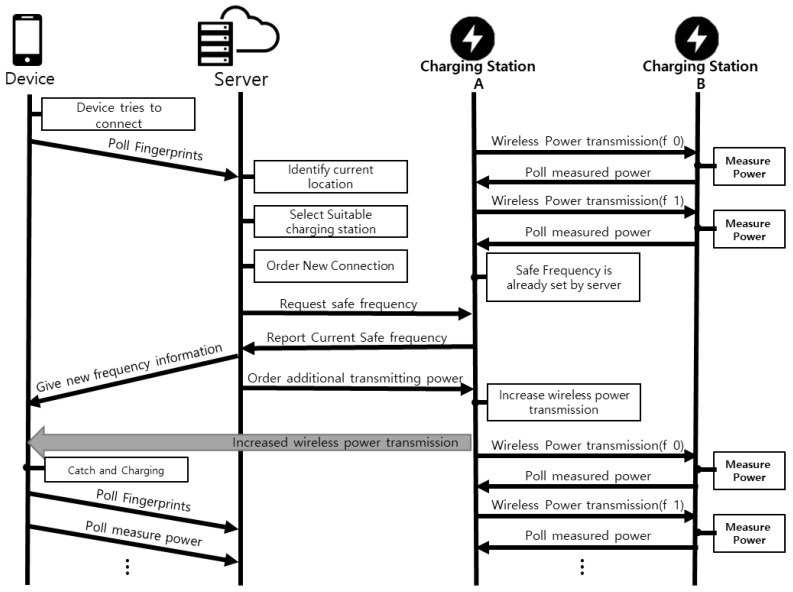
Relationships between the device, server, and charging station for a normal cycle.

**Figure 6 sensors-24-02571-f006:**
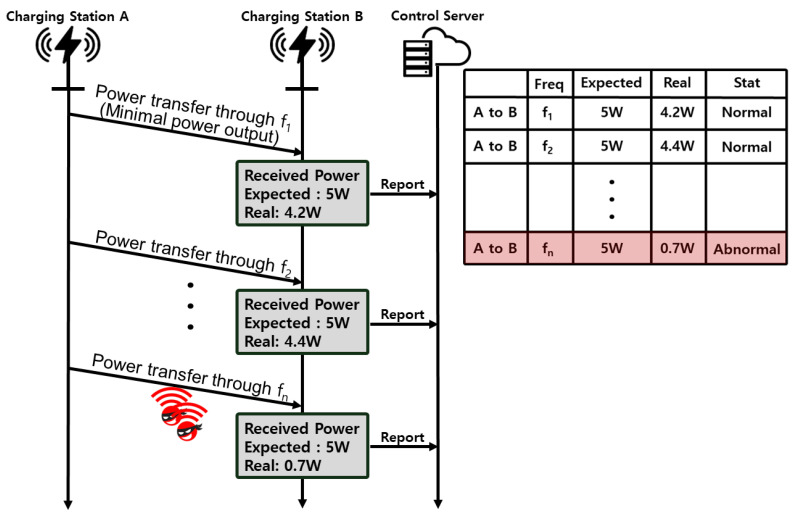
Overview of Patrol Mechanism Operation.

**Figure 7 sensors-24-02571-f007:**
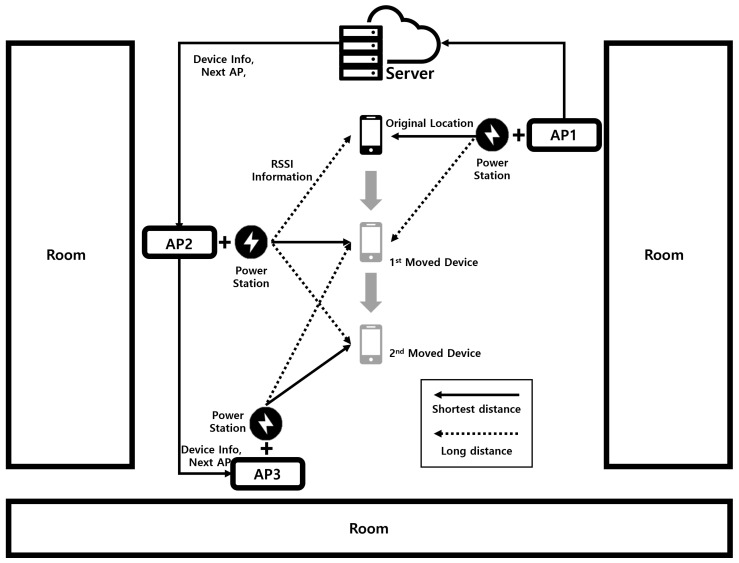
Overview of Ap Allocation Mechanism Operation.

**Figure 8 sensors-24-02571-f008:**
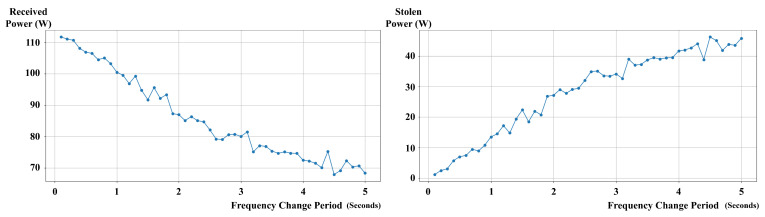
Average Received Power Across Frequency Change Intervals in the Traditional Mechanism.

**Figure 9 sensors-24-02571-f009:**
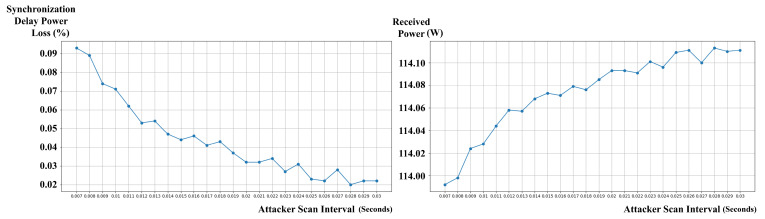
Synchronization Delay Loss at Various Attacker’s Scan Intervals in the *LazyFrog* Mechanism.

**Table 1 sensors-24-02571-t001:** Simulation Variables and Ranges.

Variable	Range	Description
FREQUENCY_RANGE	5–30 kHz	Range of frequencies used for power transmission
FREQUENCY_CHANGE_PERIOD	0.1–5 s	Time interval for frequency changes (traditional)
ATTACK_SCAN_INTERVALS	0.007–0.03 s	Time interval between attacker scans
POWER_TRANSMISSION_RATE	5–15 W	Rate of power transmission by charging stations
SIMULATION_SIZE	10×10 to 50×50	Dimensions of the virtual simulation space

**Table 2 sensors-24-02571-t002:** ANOVA results for power transmission efficiency comparison.

Mechanism	F-Statistic	*p*-Value	Effect Size ($\eta^{2}$)
Traditional vs. *LazyFrog*	F(1, 98) = 2076.28	*p* < 0.001	0.95

**Table 3 sensors-24-02571-t003:** Impact of parameters on power transmission efficiency.

Mechanism	Parameter	F-Statistic	*p*-Value	Effect Size ($\eta^{2}$)
Traditional	Frequency change period ($T_{\mathrm{fc}}$)	F(49, 450) = 283.87	*p* < 0.001	0.97
*LazyFrog*	Attack scan interval ($T_{\mathrm{scan}}$)	F(23, 216) = 214.09	*p* < 0.001	0.96

**Table 4 sensors-24-02571-t004:** Impact of frequency change period and scan interval on the attacker stolen power and the number of successful attacks.

Mechanism	Traditional		*LazyFrog*	
Parameter	Frequency change period ($T_{\mathrm{fc}}$)		Attack scan interval ($T_{\mathrm{scan}}$)	
Dependent Variable	Attacker stolen power	Number of successful attacks	Attacker stolen power	Number of successful attacks
F-statistic	F(49, 450) = 285.27	F(49, 450) = 5.01	F(23, 216) = 83.14	F(23, 216) = 216.39
*p*-value	*p* < 0.001	*p* < 0.001	*p* < 0.001	*p* < 0.001
Effect size ($\eta^{2}$)	0.97	0.35	0.90	0.96

**Table 5 sensors-24-02571-t005:** Variables and their ranges used in the sensitivity analysis.

Variable	Description	Range
Frequency change period	Time interval between frequency changes in the traditional mechanism	0.1 to 5 s
Attack scan intervals	Time interval between each scan performed by the attacker	0.007 to 0.03 s
Power transmission rate	Rate at which power is transmitted by the charging station	5 to 15 W
Simulation size	Dimensions of the virtual space in which the simulation is conducted	(10,10) to (50,50) units

## Data Availability

The original contributions presented in this study are included in the article. Further inquiries can be directed to the corresponding author.

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
