# Peer review of "LazyFrog: Advancing Security and Efficiency in Commercial Wireless Charging with Adaptive Frequency Hopping"

_sensors, 2024, doi:10.3390/s24082571_

Round 1

Reviewer 1 Report

Comments and Suggestions for Authors

Upon thorough evaluation of your manuscript titled "LazyFrog Mechanism: Enhancing Efficiency and Security in Wireless Charging Systems," I regret to convey that I cannot recommend it for publication in its current form. While your study addresses pertinent challenges in wireless charging technology, several significant concerns have been identified during the review process:

- While your study introduces the LazyFrog mechanism as a solution to enhance efficiency and security in wireless charging systems, there is a lack of clarity regarding the methodology employed to validate its effectiveness. The absence of a robust validation process undermines the credibility of the proposed mechanism and raises doubts about its real-world applicability.

- The experimental design lacks essential controls and fails to address potential confounding variables adequately. Without proper controls, it becomes challenging to ascertain whether the observed outcomes are solely attributed to the LazyFrog mechanism. Enhancing the experimental design with appropriate controls is imperative to ensure the internal validity of your study.

- The statistical analysis provided in your manuscript is inadequate in supporting the conclusions drawn. A more rigorous statistical approach, along with clear interpretation of the results, is necessary to establish the significance of the LazyFrog mechanism in enhancing efficiency and security in wireless charging systems.

- The discussion section does not thoroughly analyze the results obtained from your study. A comprehensive discussion that delves into observed trends, potential implications, and limitations of the LazyFrog mechanism is essential to provide readers with a deeper understanding of the research findings.

- The manuscript lacks sensitivity analysis to assess the robustness of the LazyFrog mechanism under varying conditions. Conducting sensitivity analyses is crucial to evaluate the mechanism's performance across different scenarios and ensure the reliability and generalizability of the findings.

In conclusion, while your study addresses an important aspect of wireless charging technology, the aforementioned concerns regarding methodology, experimental design, statistical analysis, and interpretation of results significantly impact the suitability of your manuscript for publication in our journal. I encourage you to address these issues comprehensively and consider revising your manuscript accordingly for future submission

Comments on the Quality of English Language

While the overall clarity of the writing is commendable, there are several instances of grammatical errors, awkward phrasing, and ambiguous expressions that detract from the readability and professional quality of the paper. For instance, in several sections, sentence structures are convoluted, making it challenging for readers to grasp the intended meaning easily. Additionally, there are instances where incorrect verb tense usage and inconsistent formatting of technical terms are observed, further hindering comprehension.

Author Response

  • Manuscript ID: sensors-2884162
  • Manuscript Title (Maintained): “LazyFrog: Advancing Security and Efficiency of Commercial Wireless Charging with Adaptive Frequency Hopping”
  • Authors: SungKyu Ahn, HyeLim Jung and Ki-Woong Park

March 25, 2024

Dear Reviewers of this manuscript:

We wish to submit our revised manuscript reflecting the reviewers’ comments regarding publication in Intelligent Sensing Techniques for Detection of Attacks against Public Infrastructure, a special issue of Sensors, titled “LazyFrog: Advancing Security and Efficiency of Commercial Wireless Charging with Adaptive Frequency Hopping”

We would like to thank the reviewers for their time and valuable comments. We thoroughly revised the manuscript in accordance with your comments, and our responses are enclosed. All of your comments were helpful toward improving the quality of this paper.

Thank you.

Sincerely,

Prof. Dr. Ki-Woong Park (Corresponding Author)

Reviewer 2 Report

Comments and Suggestions for Authors

some points require to be addressed:

1.     The current literature review is not sufficient. It is better to use newer references (2023-2024) and update the introduction section.

2.    There seems no link between the literature review presented and the current work.

3-    Description of Figure 1 should be added in section 2 of the manuscript.

4-    The structure of the article is very good. I recommend further discussion about the proposed methodology.

5-    The objective and methodology should be better described and justified.

6-    Proofread the entire manuscript to rectify some existing typos and grammatical errors.

Comments on the Quality of English Language

some points require to be addressed:

1.     The current literature review is not sufficient. It is better to use newer references (2023-2024) and update the introduction section.

2.    There seems no link between the literature review presented and the current work.

3-    Description of Figure 1 should be added in section 2 of the manuscript.

4-    The structure of the article is very good. I recommend further discussion about the proposed methodology.

5-    The objective and methodology should be better described and justified.

6-    Proofread the entire manuscript to rectify some existing typos and grammatical errors.

Author Response

(The authors gave the same response as above.)

Reviewer 3 Report

Comments and Suggestions for Authors

The article presents a mechanism named LazyFrog to improve the charging efficiency and convenience of wireless charging technology. The contents are suitable for the scope of the journal and can attract sufficient interest from the readers. However, further improvements are still needed before its final publication, and I have decided to give a 'minor revision', 

My comments are as follows:

1. The introduction of the article requires improvement, and it is suggested that the content of Section II be integrated into the introduction to provide a more cohesive and comprehensive overview of the topic.

2. The authors' explanation of how the attacker could impact the charging of other normal devices is not sufficiently clear. Power leakage damages the interoperability (https://doi.org/10.3390/en16041653) of normal wireless charging devices, thereby affecting the charging of other devices and leading to critical operational problems.

3. Figure 4 depicts only the foggy server and mobile device. It is recommended to include charging stations in the figure to provide a more comprehensive representation of the system.

4. Figure 8 should include labels for the x-axis and y-axis to enhance clarity and facilitate understanding.

5. It is important to note that the actual efficiency of wireless charging cannot reach 100%. The author should focus on explaining the evaluation criteria used to assess efficiency (https://doi.org/10.1007/978-3-642-33741-3_14).

6. Line 157, 'power transmission efficiency was further improved by the magnetic coupler'. It should be noted that the materials improvement/modification of the magnetic coupler can help enhance the efficiency, and some introductions on this aspect should be further supplied. For example, nanocrystalline alloy film for wireless charging systems, and high specific strength and high specific modulus properties, such as fiberglass.

Author Response

(The authors gave the same response as above.)

Reviewer 4 Report

Comments and Suggestions for Authors

In this paper, an improved mechanism had been proposed for facilitating secure wireless charging to authorized users while attempting to minimize access to such services to malicious and unauthorized users. The proposed mechanism is based on establishing a pseudo-random frequency hopping pattern for delivering the wireless charge, whereby the frequency hopping is only done when a malicious access is detected. In general, the details of the proposed mechanism had been described quite well in the article and appears to offer a promising improvement compared to existing schemes.

Section 4 further highlights the merits of the proposed mechanism and the added performance gain. However, there a are a few details in that section that are either missing or needs to be further explained. First, the details of the simulation setup are incomplete and prevents an interested reader with reproducing and/or verifying the same results presented in the manuscript. Furthermore, can you please clarify the details of the “Traditional Wireless Power Transmission Model” that was used to contrast with the performance of the proposed wireless charging mechanism? Is it one of the mechanisms referred to in the related works? If not, then I would recommend using them for a better comparison. Moreover, the results in tables 1 and 2 would be best shown together and in a graphical form for a better highlight of the differences in performance, and the graph in Figure 8 are missing the labels on their axes.

Author Response

(The authors gave the same response as above.)
